# Social distancing compliance: A video observational analysis

**Evelien M. Hoeben**[1]*, **Wim Bernasco**[1,2], **Lasse Suonperä Liebst**[3], **Carlijn van Baak**[1], **Marie Rosenkrantz Lindegaard**[1,2,4]

**1** Netherlands Institute for the Study of Crime and Law Enforcement (NSCR), Amsterdam, The Netherlands, **2** Department of Spatial Economics, School of Business and Economics, VU University Amsterdam, Amsterdam, The Netherlands, **3** Department of Sociology, University of Copenhagen, Copenhagen K, Denmark, **4** Department of Sociology, Faculty of Social and Behavioural Sciences, University of Amsterdam, Amsterdam, The Netherlands

* ehoeben@nscr.nl

**Data Availability Statement:** The data on social distancing regulations and the COVID-19 transmission in the Netherlands from the National Institute for Public Health and the Environment

## Abstract

### Purpose

Virus epidemics may be mitigated if people comply with directives to stay at home and keep their distance from strangers in public. As such, there is a public health interest in social distancing compliance. The available evidence on distancing practices in public space is limited, however, by the lack of observational data. Here, we apply video observation as a method to examine to what extent members of the public comply with social distancing directives.

### Data

Closed Circuit Television (CCTV) footage of interactions in public was collected in inner-city Amsterdam, the Netherlands. From the footage, we observed instances of people violating the 1.5-meter distance directives in the weeks before, during, and after these directives were introduced to mitigate the COVID-19 pandemic.

### Results

We find that people complied with the 1.5-meter distance directives when these directives were first introduced, but that the level of compliance started to decline soon after. We also find that violation of the 1.5-meter distance directives is strongly associated with the number of people observed on the street and with non-compliance to stay-at-home directives, operationalized with large-scale aggregated location data from cell phones. All three measures correlate to a varying extent with temporal patterns in the transmission of the COVID-19 virus, temperature, COVID-19 related Google search queries, and media attention to the topic.

### Conclusion

Compliance with 1.5 meter distance directives is short-lived and coincides with the number of people on the street and with compliance to stay-at-home directives. Potential

(RIVM; tinyurl.com/ya9yywoj), the data on temperatures from the Royal Netherlands Meteorological Institute (KNMI; tinyurl.com/yb2mjwaw), the data on internet search queries from Google Trends (trends.google.com), and the data from Google's COVID-19 Community Mobility Reports (google.com/covid19/mobility/) are publicly available without restrictions. The media data from the ANP have been deposited to OSF (osf.io/59tnu). Access to the raw CCTV footage data will be granted by signing an agreement stating that the applicant (1) will use the data only for scientific purposes, (2) will not make the data accessible to third parties, and (3) will not publish results that will disclose the identity of the subjects in the data. To request access to the raw footage files or inquire about the conditions, please contact Thomas Hoogeboom, datamanager at the Netherlands Institute for the Study of Crime and Law Enforcement (NSCR), at email address nscr@nscr.nl. The analyzed data file with coded observations of the CCTV clips has been deposited to OSF (osf.io/59tnu). DOI of all deposited data for this project: 10.17605/OSF.IO/59TNU. These datasets allow readers to replicate the analytical parts of our research.

**Funding:** This project was supported by the Netherlands Institute for the Study of Crime and Law Enforcement (NSCR, www.nscr.nl) and by a grant from The Netherlands Organisation for Health Research and Development (ZonMw, www.zonmw.nl; project nr. 10430022010017), awarded to MRL, WB, and EH. NSCR and ZonMw had no role in study design, data collection and analysis, decision to publish, or preparation of the manuscript.

**Competing interests:** The authors have declared that no competing interests exist.

implications of these findings are that keep-distance directives may work best in combination with stay-at-home directives and place-specific crowd-control strategies, and that the number of people on the street and community-wide mobility as captured with cell phone data offer easily measurable proxies for the extent to which people keep sufficient physical distance from others at specific times and locations.

## Introduction

Social distancing has been a critical non-pharmaceutical measure to slow the spread of the COVID-19 coronavirus. The World Health Organization recommends social distancing as a most effective anti-pandemic measure [1]. Historical pandemics are a case in point, including the Spanish 1918 flu pandemic, in which cities implementing social distancing faced lower death rates [2]. Similarly, in the ongoing COVID-19 pandemic, geographical differences in transmission rates and death rates appear to be linked to how firmly social distancing is implemented [3–5]. While the effectiveness of social distancing hinges on the cooperation of the public [6], it may be challenged by the human compulsion for engaging in proximate face-to-face interactions [7]. Therefore, policy-makers need evidence of social distancing compliance to inform policy decisions on whether to implement rules on a voluntary or compulsory basis. So far, however, little is known about the extent to which people keep physical distance from others in everyday practice, not only in the ongoing crisis but during pandemics in general.

The current study aims to provide empirical evidence of compliance with social distancing directives in the weeks after their introduction. Particularly, the study focuses on the extent to which people keep 1.5 meter physical distance from strangers in public space. Of the various social distancing practices (e.g., staying at home, preventive quarantine, avoiding crowds), keeping a physical distance from others is the most directly related to disease transmission. After all, exposure to the virus occurs if people are in physical proximity to a COVID-19 positive individual. Other social distancing measures, such as avoiding non-essential trips outdoors, will only indirectly affect the risk of transmission, because they prevent or reduce the likelihood of physical proximity between people. Yet, with some exceptions [8, 9], the extant research on social distancing has not been able to directly evaluate whether people keep their distance. Instead, a growing body of work relies on aggregated location data from cell phones to determine adherence to stay-at-home directives [10–14] or on people's self-reports about their willingness to comply with a wide variety of social distancing directives [15–17]. Neither of these approaches are suitable to assess whether people comply to 1.5 meter distance guidelines, because they are insufficiently accurate in determining proximity or because they do not provide objective information on people's behavior. Exceptions are studies that apply Bluetooth technology [9], pedestrian tracking sensors [8] or other methodology to precisely pinpoint the proximity between people.

The current study relies on observations of real-life behavior in public spaces, as captured by Closed Circuit Television (CCTV) clips. Real-life behavioral data of this kind provide ecologically valid evidence on variation in social distancing non-compliance [18]. Furthermore, our observational approach addresses a call within epidemiology for quantitative research on disease-behavior interactions [6]. Specifically, real-life behavioral observation is, as stressed by Verelst et al. "desperately needed" [19 p. 12] for the validation and parametrization of disease simulation models, which have been instrumental in shaping policy responses to the coronavirus pandemic.

Scholars have explicitly called for research describing how individuals change their everyday behavior during pandemics and in the face of the threat of disease [6, 20]. Evidence on compliance during the COVID-19 pandemic [21], society-wide non-infectious emergencies (e.g., terrorist attacks; [22]) and medical adherence to long-term treatment [23] shows that people's responses tend to go through transient phases—with major initial changes in activities being followed by a gradual return to everyday routines. Based on this earlier work, we expect to find that people will comply with regulations in the immediate period after such regulations have been introduced, but that their compliance will decrease gradually in the following period.

In addition to examining temporal variation in social distancing compliance, we also investigate factors that might explain such variation. First, we examine the role of a purely situational constraint: the number of people present on the street. As the number of people on the street increases, it may simply be more difficult for people to keep the described distance from others. The number of people on the street may also have a psychological effect on people: Relatively empty areas provide a visual reminder that a crisis is at hand, which might affect people's awareness of their own behavior and that of others [24, 25]. Second, we expect that patterns in compliance to keep-distance directives will co-occur with community-wide adherence to stay-at-home directives, as indicated by large-scale cell phone data on the time that people spent at non-residential locations. This measure not only reflects temporal variation in general compliance to governmental regulations, but it is expected to also affect the number of people on the street and, thereby, the likelihood of social distancing violations. Third, we expect that non-compliance will align with the increasing temperatures that mark the start of Spring in The Netherlands, since improved weather conditions allow for more outdoor socializing [26]. Thus, we expect that temperatures affect the number of people on the street and, in turn, people's compliance to keep-distance directives. Fourth, we expect that non-compliance trends may unfold as the perceived urgency of the problem decreases, as indicated by the number of registered transmissions and deaths. Survey research shows that people's willingness to comply is strongly driven by fear and anxiety related to the virus [27]. Fifth, we expect that patterns of non-compliance can be explained with collective attention to the COVID-19 pandemic. When a dramatic incident occurs, all members of the collective are emotionally affected. Such shared trauma leads to feelings of connectedness and stimulates cohesion [22]. In this context, compliance to guidelines for collective well-being, such as preventive social distancing directives, is likely to be socially rewarded, whereas non-compliance is likely to be met with strong disapproval or even hostility. As the collective attention wears off, the potential social benefits of compliance decrease, leading people to return to their routines. Generally, we expect compliance with social distancing to be positively related to the salience of COVID-19 related concerns in society, as reflected by the content of traditional media and topic searches on the internet.

## Materials and methods

Our data on social distancing compliance are derived from CCTV clips of public behavior, captured by municipal surveillance cameras in inner-city Amsterdam, the Netherlands. These recordings were collected over the course of 10 weeks during the COVID-19 outbreak, from February 29th to May 2nd 2020, and were provided by the Amsterdam police and municipality. Access was provided under the condition that the data would be securely stored, not be publicly shared, and that the identity of the individuals visible in the footage would be protected. The project has been approved by the Ministry of Internal Affairs (PaG/BJZ/49986). We do not have personal informed consent from the persons recorded and it is not practically

possible to obtain such consent, nor would it be ethically responsible for us to attempt to obtain such consent. However, the Ministry of Internal Affairs provided the consent to use CCTV footage on behalf of the individuals recorded by the surveillance cameras in public. The project as well as this consent procedure have been approved by the Ethics Committee for Legal and Criminological Research (CERCO) at the Vrije Universiteit Amsterdam.

Recordings were sampled from 3 cameras, located in relatively busy areas in the city near the central train station and around shopping areas. We sampled one weekday (Thursday) and one weekend day (Saturday) per week. To assess the potential impact of the social distancing directives, we included the weeks during which the social distancing directives were announced as well as the weeks before and after this period. Due to technical issues in 2 of the 10 weeks, the footage was available for only one of the selected days. In total, we included footage from 18 days, of which we selected the 5-minute period from 16:00 to 16:05 (4:00 pm to 4:05 pm). This time slot was chosen because it reflected a busy time of day (e.g., opening hours of the shopping areas and commuting around the central station). The resulting 54 5-minute clips are a subset of the more than 20,000 hours of recordings captured by 55 surveillance cameras. We coded only a small subset of the available data, because manual coding of video behavioral data is very labor-intensive [28].

The 5-minute clips were coded for instances where people were within 1.5 meters proximity of each other or formed a group of more than three individuals. These behavioral definitions align with the official Dutch social distancing regulations [29]. The coder played back the video (if necessary in slow motion and repeatedly) and registered every social distancing violance observed, marking the time point. We acknowledge that our distance measures are not perfect. In particular, human judgement of distance can be affected by the focal length of the camera. With increasing focal length (zooming out) the same distance appears larger. However, all coded CCTV clips were recorded in dense urban environments where, throughout the camera viewshed, various objects were available to serve as reliable reference objects for length estimation, including sidewalk tiles, benches, trash cans, and bikes.

For each observed social distance violation (i.e., 1.5-meter proximity and group formation), the coder judged whether the people involved were members from the same household and, as such, were exempt from the social distancing regulation. People were considered to belong to the same household when they were observed to make physical contact (e.g., holding hands). This assessment was based on evidence suggesting that relationship ties may be inferred from visual behavioral information (known as "tie-signs" [24, 30, 31]). In the current study, we only present social distancing violations that were presumably by non-family members. We established a correlation of 0.99 between the measures corrected and uncorrected for household membership. Analyses (trends, de-trended bivariate correlations, and trivariate regression models) with counts uncorrected for household membership showed similar findings.

The CCTV-clips were also used for counting the number of people present on the street. Since here we were not interested in specific behavior but in the average number of people present over the 5-minute sample, we used a slightly different coding procedure. While playing the 5-minute clips, the video was halted 5 times (at 16:00, 16:01, 16:02, 16:03, and 16:04) and the number of people in the still picture was counted. The measure of the number of people on the street is the mean of these five measurements per clip. For the coding of the number of people on the street as well as of the social distancing violations, two researchers observed and discussed a subset of the clips together, discussed agreements, and altered operational definitions of behaviors accordingly [32]. The resulting codebook is provided in the S1 Appendix.

We made minor corrections to the observed data due to incidental changes in camera zoom levels. On three days, one of the cameras recorded with a higher zoom level than on other days, which affected the amount of space covered in the viewshed and, thereby, reduced

the number of violations that could be detected. In an attempt to correct for this bias, on April 4[th], we replaced the 16:00 (4 pm) observations by observations from 13:00 (1 pm). On March 26[th] and March 28[th], one camera was zoomed in all day. The observations from that one camera for those days were discarded and imputed with OLS-based predicted values based on the observed values of the other two cameras. The regression equation across the whole study period (all dates except March 26[th] and March 28[th]) has an explained variance ($R^2$) of 0.73. This procedure increased the number of violations on March 26[th] from 80 to 90 and on March 28[th] from 79 to 114. The same approach was used to correct the measure of the number of people on the street. The explained variance of the respective regression equation ($R^2$) was 0.52. The Figures with the corrected measurements are provided and discussed in the main text. The Figures with the uncorrected unimputed measurements are presented in the S2 Appendix. The Figures show similar patterns of social distancing violations over time.

To contextualize the temporal patterns of observed social distancing violations, we collected information from various sources. Except for the media data, these sources are publicly available without restrictions. (1) We constructed a timeline of the implementation of the social distancing regulations in The Netherlands based on official information of the government [33] as well as timelines published in news media [34, 35]. (2) We collected records of the COVID-19 transmission in The Netherlands, as expressed by the absolute number of deaths and the absolute number of new transmissions over time. This information was registered by the National Institute for Public Health and the Environment (RIVM) and included in the global database of John Hopkins University. The data were downloaded on May 3[rd] from tinyurl.com/ya9yywoj [36]. (3) Compliance to stay-at-home directives was operationalized with data from Google's COVID-19 Community Mobility Reports, which is based on cell phone location and movement information collected from users who opted-in to Location History for their Google Account. These data capture relative changes in the number of visits and length of stay at (a) parks, (b) retail and recreation, (c) grocery and pharmacy, (d) transit stations, and (e) workplaces. The data represent changes compared to a baseline, which is the median value for the corresponding day of the week during the 5-week period from January 3[rd] to February 6[th] in 2020. Because the baseline is specific to the day of the week, these data have already been corrected for weekly periodicity (unlike the data from other sources that we apply). We used data specific to Amsterdam and created a combination score that is the average percentage change across the five categories of non-residential places. The data were downloaded on August 31[st] from Google [37]. (4) Daily temperatures at 16:00 (4 pm) were obtained from the hourly weather database of the Royal Netherlands Meteorological Institute (KNMI). We used measures from weather station Schiphol Airport, located a few kilometers south of Amsterdam. The data were downloaded on May 5[th] from tinyurl.com/yb2mjwaw [38]https://projects.knmi.nl/klimatologie/uurgegevens/selectie.cgi. (5) Collective attention to the pandemic was assessed with two measures. First, internet search queries in The Netherlands were captured with Google Trend data for the search terms 'corona' and 'COVID-19' (downloaded on May 6[th] from trends.google.com [39]). Google Trend offers an index of how widely search queries are used in certain regions—with a score of 100 representing the peak popularity for a search term and a score of 50 indexing half the popularity. Second, media attention was captured with data from the largest news agency company in The Netherlands, Algemeen Nederlands Persbureau (ANP). This agency provides various news media channels with fact-checked information, among which newspapers, radio, and television. Data include the number of messages from their newsstream Medianet and topic-specific newsfeed Buzz that include the terms 'COVID-19,' 'corona,' and derivatives of the term 'corona' (e.g., coronavirus).

The covariation between the number of social distancing violations, the number of people on the street, compliance with stay-at-home-directives (Google mobility), temperature, COVID-19 transmissions and deaths, Google search scores, and media items was further examined by analyzing bivariate correlations and trivariate regression models. When examining covariation among trended time series, one needs to account for autocorrelation (i.e., correlation between values within the same variable that are $k$ time periods apart) and for common dependence on time [40]. For more information about the extent of the autocorrelation and cross-correlations among the investigated variables (i.e., correlations between a measure of variable X at time $t+k$ with $k$ being the lag and a measure of variable Y at time $t$), see the S3 Appendix. Please note that the bivariate correlations and trivariate regressions are used to examine contemporaneous effects.

To examine covariation among the time series, we de-trended the variables prior to the calculation of the bivariate correlations and the trivariate regression models. The ten variables in the analysis do not necessarily follow a common trend, and some are likely to display weekly periodicity in addition to a linear or curvilinear trend. In line with the approach advocated in the literature on estimating relationships between time series [41], trend and periodicity were removed from each of the variables separately by regressing them on time (days since start of the series), time squared, and a dummy variable indicating whether the measure was from a Saturday as opposed to from a Thursday (to account for weekly periodicity). The residuals of these models represent their variation with trend and periodicity removed. Upon request by one of the reviewers, for the purpose of providing a complete description of the relations between the variables before detrending, we present the bivariate correlations and trivariate regression models as calculated with the raw data in S4 Appendix. The details of the detrending procedure are described in S5 Appendix.

All analyses were executed with version 4.0.0 of the R package for Statistical Computing [42]. The coding of the CCTV-clips was done in Microsoft Excel. The R code and the data files with the coded observations of the CCTV-clips are accessible through an OSF depository (osf.io/59tnu).

## Results

The first case of COVID-19 in The Netherlands was confirmed on February 27[th] 2020 and case number 100 was confirmed 11 days later. The virus progressed rapidly through the community, peaked mid-April, and then declined—as indicated by the number of new transmissions and related deaths presented in Fig 1A and 1B.

In response to the outbreak, the Dutch government implemented a series of social distancing regulations—as summarized in the timeline, Fig 2. The outbreak in the Netherlands started in the province of North-Brabant and, therefore, the first regulations were focused on this subregion (March 6[th]). Directives for the entire country soon followed (March 9[th], March 12[th]). On March 15[th], the government first mentioned 1.5 meter as a reference distance [43]. The official lockdown was announced in a press conference on March 23[rd]. In addition to previous directives, the government introduced fines for the violation of the social distancing regulations. In this press conference, the government restated the explicit rule of keeping 1.5-meter distance and also prohibited group gatherings, defined as the co-presence of three or more people who do not keep a 1.5-meter distance from each other. This included coincidental group gatherings. Members from the same households and children under the age of 12 were exempted from these rules [29].

### Compliance to 1.5-meter distance and stay-at-home directives over time

Fig 3 displays the number of observed social distancing violations (i.e., < 1.5-meter proximity by people not sharing the same household) from February 29[th] to May 2[nd]. Note that we refer

**Fig 1. Covariates over time.** The panels display time trends in (A) COVID-19 deaths, (B) COVID-19 transmissions, (C, D) media items, (E, F) Google search scores, (G) compliance with stay-at-home-directives (Google mobility), (H) temperature, and (I) the number of people on the street.

to 'social distancing violations' throughout, even though the period covered includes weeks before the social distancing directives were made explicit and before their violation was legally enforced.

As can be seen in Fig 3, there is a decline in the number of violations between March 7th and March 21st, which coincides with the first announcements of social distancing directives on March 9th, March 12th, and March 15th. Note that these announcements regarded general social distancing directives (e.g., avoid shaking hands, work from home), but that the

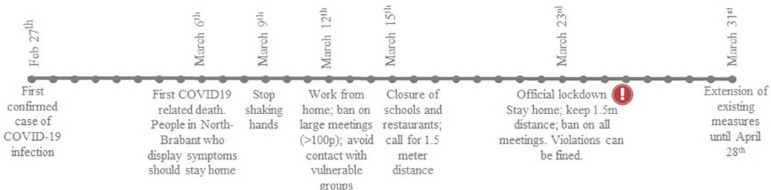

**Fig 2. Timeline of the COVID-19 outbreak and the implementation of social distancing directives in The Netherlands.**

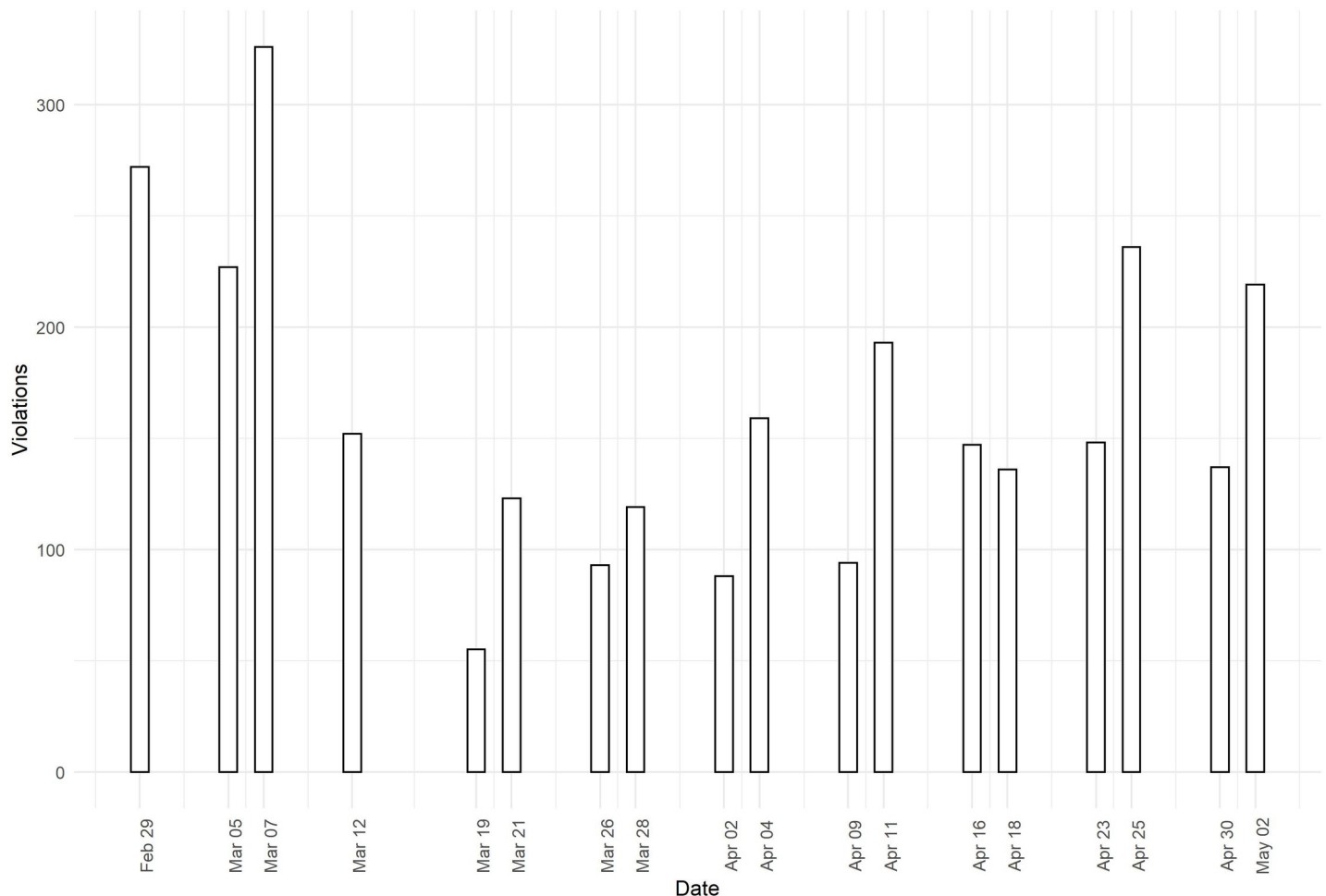

**Fig 3. Observed social distancing violations (i.e., < 1.5 meter proximity by non-household members) from CCTV clips.**

1.5-meter distance directive was not mentioned until March 15th and not sanctioned until March 23rd. On March 23rd, the government announced explicit rules related to keeping 1.5-meter distance and avoiding crowding (> 3 people) in public spaces. On this date, the government also installed fines for rule violators. The number of violations appeared to be lowest between March 19th and April 2nd. From April 2nd onwards, a steady increase in the number of social distancing violations is visible, especially on the weekend days (April 4th, April 11th, April 18th, and April 25th are Saturdays).

To statistically examine the time trend in the number of social distancing violations, we conducted Ordinary Least Squares regression models with the number of violations regressed on linear time, quadratic time, and periodicity due to the data being collected on Thursdays and Saturdays. The outcomes of these models are presented in Table 1. The findings show that the linear model does not provide a good fit to the data (i.e., the coefficient for linear time is not significant, $R^2 = 0.05$), that the quadratic model provides a much better fit (i.e., the coefficients for time and time squared are significant and the $R^2$ of 0.53 is much higher compared to that of the linear model), and that the quadratic model that includes a control for periodicity fits even better (i.e., the coefficient of the dummy variable for weekday is significant, the $R^2$ is 0.72). The predicted values from these three models are plotted against the observed values in

**Table 1. Number of social distancing violations regressed on linear time, quadratic time, and periodicity.**

| | Model 1: Linear | Model 2: Quadratic | Model 3: Quadratic + periodicity |
|---|---|---|---|
| Intercept | 188.71*** | 280.65*** | 243.44*** |
| | (33.35) | (33.91) | (29.57) |
| Time | -0.79 | -9.77*** | -9.06*** |
| | (0.87) | (2.40) | (1.92) |
| Time squared | | 0.14** | 0.13*** |
| | | (0.04) | (0.03) |
| Saturday (0/1) | | | 60.65** |
| | | | (19.55) |
| $R^2$ | 0.05 | 0.53 | 0.72 |
| N | 18 | 18 | 18 |

***$p < 0.001$

**$p < 0.01$, *$p < 0.05$.

Estimates from OLS regression models. Standard errors in parentheses. Time = number of days since February 29, 2020.

Fig 4. Fig 4 shows that, for both U-curves (based on the quadratic model and the quadratic model corrected for periodicity), the inflection point occurs in early April, about 1.5 week after the introduction of the social distancing directives. Thus, we detect a trend in the number of social distancing violations over time, in which this number decreased between late February and early April, but increased from early April to early May.

At first glance, this pattern of violations roughly coincides with the plateau of the spread of COVID-19 in the Netherlands; the number of new transmissions is highest from April 9[th] to 11[th] and then starts to go down (Fig 1B). The correlations between the number of violations, on the one hand, and the number of new deaths and infections, on the other hand, are visualized in Fig 5A and 5B. After correcting for trend and periodicity in the time series (i.e., detrending the data), we find a weak correlation between the number of social distancing violations and the number of new cases, and a moderate positive correlation between the number of social distancing violations and the number of new deaths (Fig 6). This means that, in contrast to what we would expect, the number of social distancing violations increases as the number of new deaths increases. This could be an indication that people's willingness to comply is not as strongly driven by fear and anxiety related to the virus as has been suggested in survey research [27]. Note, however, that we did not establish any significant effects between these factors and the number of social distancing violations after controlling for the number of people on the street (Table 2).

To measure the overall number of people on the street we used the same sample of the CCTV footage that was used to measure social distancing violations. Fig 1I shows the number of people on the street in the weeks prior to the introduction of the social distancing directives (February 29[th] to March 7[th]), the weeks in which new directives were introduced (March 12[th] to April 2[nd]), and the first weeks after their implementation (April 4[th] to May 2[nd]). There is a strong similarity in the temporal patterns of the number of social distancing violations (i.e., < 1.5-meter proximity by people not sharing the same household) and the number of people on the street. When the first social distancing directives were implemented, fewer people were visible in public space, but over time this number increased. The number of people on the street was higher on the weekend than on weekdays, which is also in line with the patterns observed

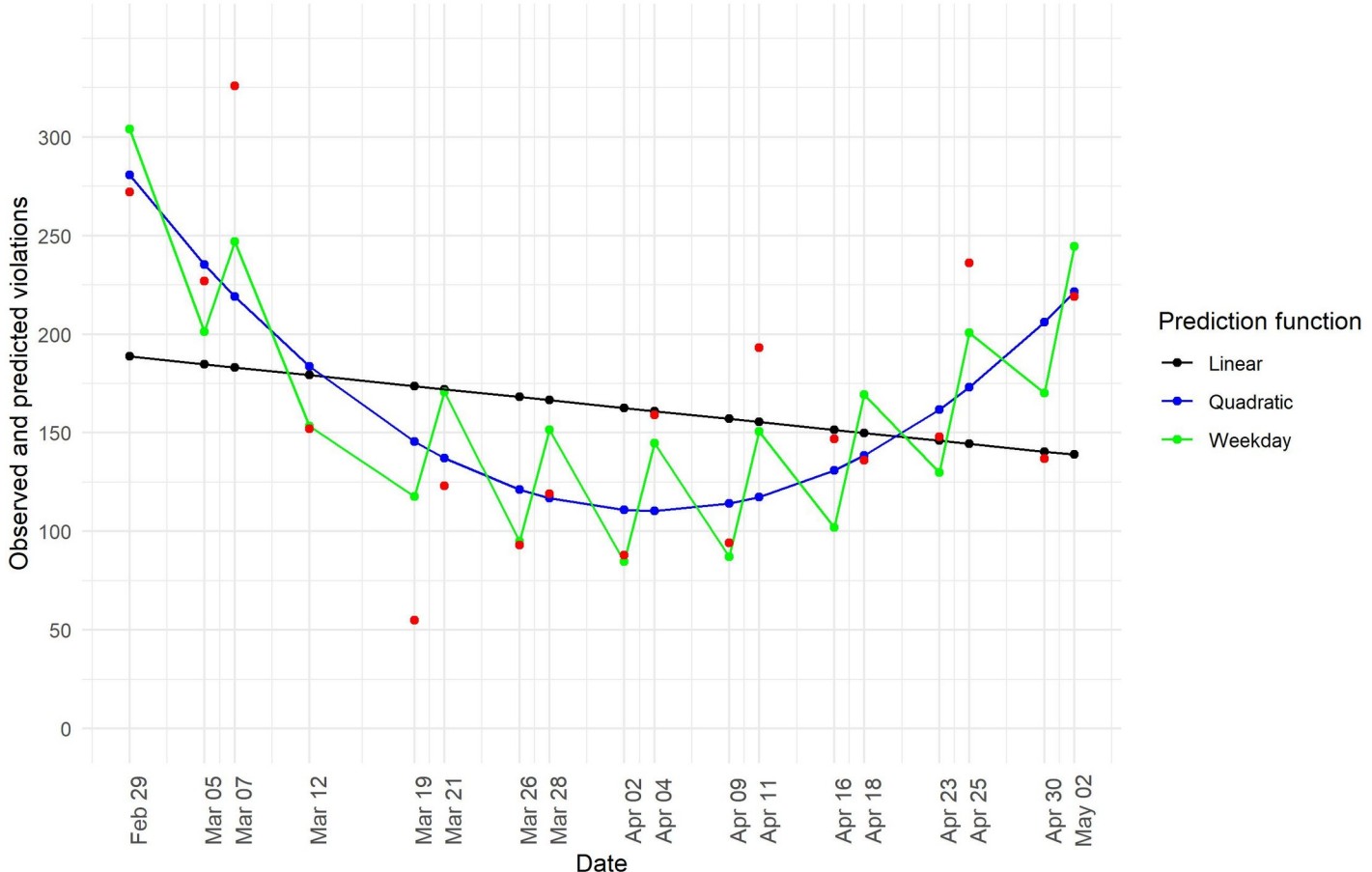

**Fig 4. Social distancing violations as a function of time.** Observed values (in red), and fitted values of a linear model (Model 1 in Table 1; in black), a quadratic model (Model 2 in Table 1; in blue), and a quadratic model that corrects for periodicity due to weekday (Model 3 in Table 1; in green).

for social distancing violations. Fig 5I displays the relationship between the number of people on the street and social distancing violations, which is positive and quite strong ($R = 0.81$, $R^2 = 0.66$ after detrending the time series, see Fig 6).

Fig 1G displays the changes in daily time spent at non-residential places (i.e., parks, retail and recreation, grocery and pharmacy, transit stations, and workplaces) in Amsterdam in the period from February 29th to May 2nd compared to a baseline value for the corresponding day of the week during the period of January 3rd to February 6th 2020. We see that, until approximately March 12th, the mobility patterns are similar to those in the preceding period (i.e., small variation around the baseline). In the week of March 12th to March 19th, there is a sharp decline visible in the time that was spent at non-residential locations. This relative time spent away from the home remained low until approximately April 4th, after which it slowly started to increase (with the exception of Easter Monday on April 13, which is a national holiday). The temporal patterns in community-wide mobility are similar to those in the number of social distancing violations, as visualized in Fig 5G and by the established bivariate correlation of 0.78 ($R^2 = 0.61$; Fig 6) with the de-trended data. Interestingly, this relationship between mobility and the number of social distancing violations remains significant after controlling for the number of people on the street (Table 2).

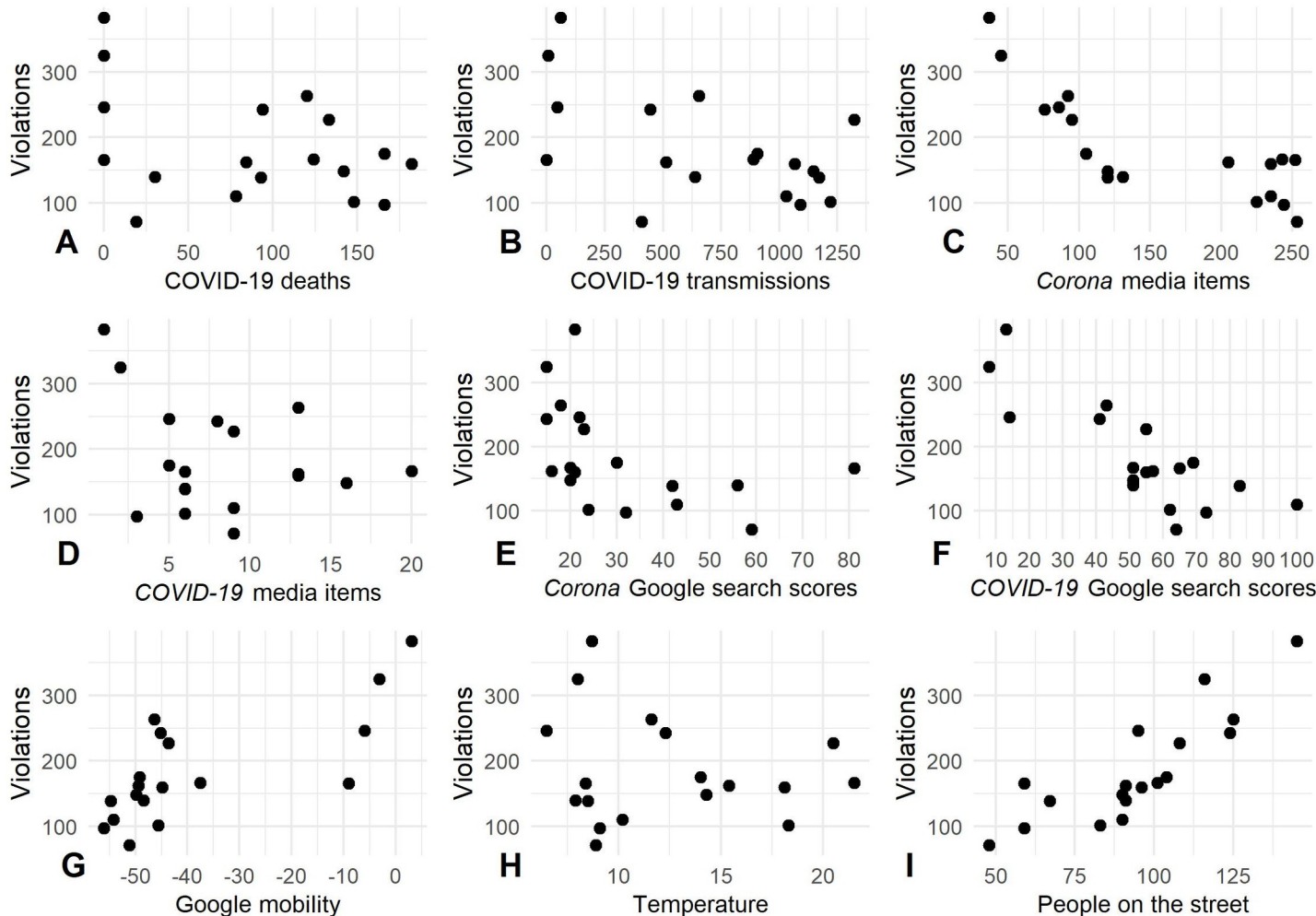

**Fig 5. Correlations between the number of observed social distancing violations and the covariates.** The panels display the correlations between, on the one hand, the number of observed social distancing violations and, on the other hand, the (A) COVID-19 deaths, (B) COVID-19 transmissions, (C, D) media items, (E, F) Google search scores, (G) compliance with stay-at-home-directives (Google mobility), (H) temperature, and (I) the number of people on the street.

## Contextualizing social distancing compliance: Temperature and collective attention

Of the covariates that we examined, social distancing violations appear most strongly related to the number of people on the street ($R = 0.81$, $R^2 = 0.66$) and mobility ($R = 0.78$, $R^2 = 0.61$). We find substantively weaker correlations with all the other variables (Fig 6, bottom row).

Specifically, we find a moderate positive correlation between the number of violations and temperature. Although we cannot establish causality, we suggest that the temperature affects the number of people that are outdoors [26], which, in turn, may affect people's ability to keep the prescribed distance. In line with this suggestion, we find that temperature does not remain a significant covariate of the number of social distancing violations after controlling for the number of people on the street (Table 2).

Further, we find a weak correlation between the number of violations and the number of 'COVID-19' media messages, and a moderate negative correlation with the number of 'Corona' media messages: As the number of media messages on this topic decreases, the number of social distancing violations increases. These covariates were not significantly related to

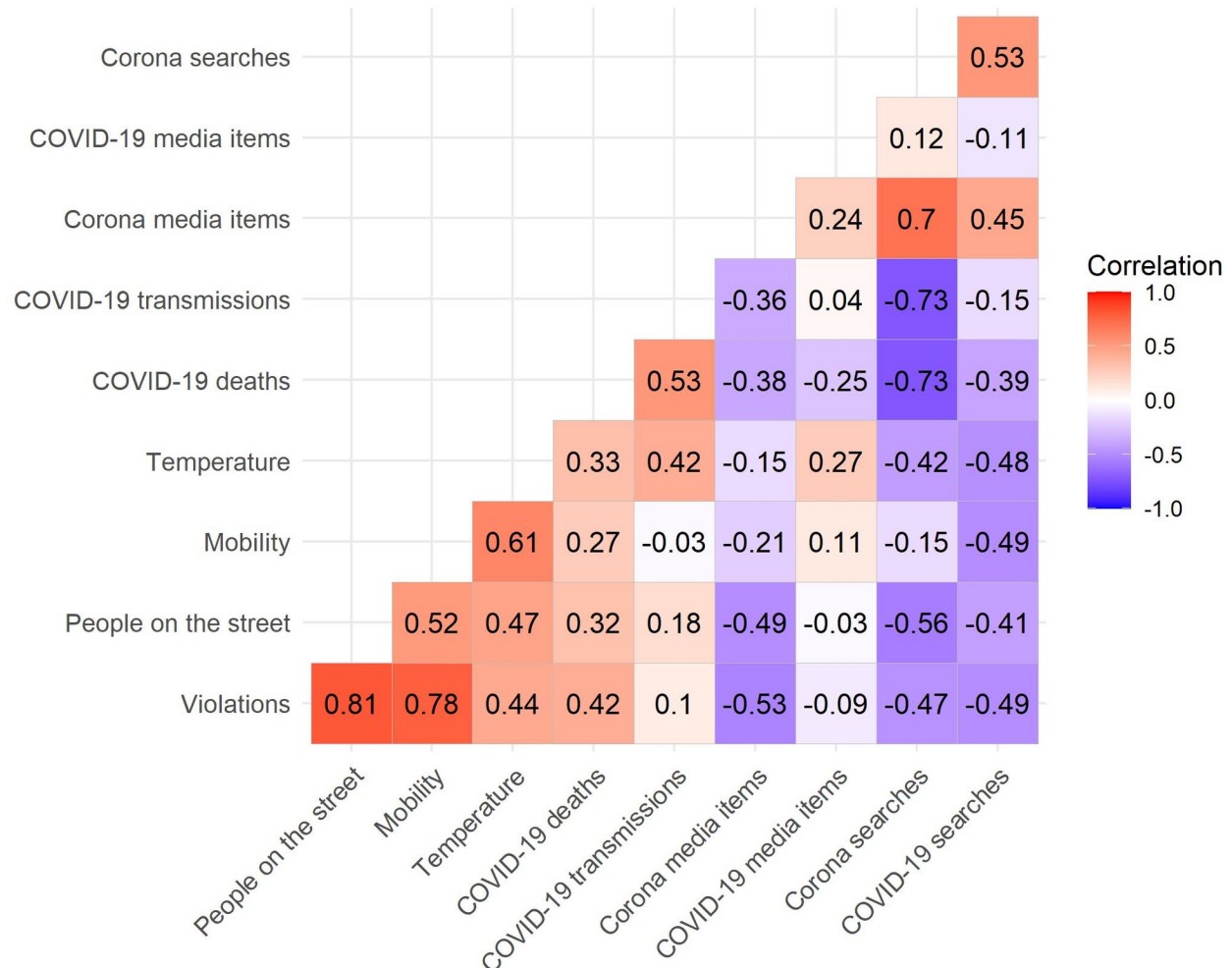

**Fig 6. Correlation coefficients between de-trended variables.** All variables were de-trended prior to including them in the analyses by taking residuals after OLS-regressing them on time, time squared, and periodicity due to weekday. Time points are Thursdays and Saturdays between Saturday February 29th, 2020 and Saturday May 2nd, 2020.

the number of social distancing violations after we accounted for the number of people on the street (Table 2).

Finally, we find moderate negative correlations between Google search behavior and the number of social distancing violations that do no longer reach conventional levels of significance after controlling for the number of people on the street (Table 2).

Table 2 shows that, after controlling for the number of people on the street, only mobility remains a significant predictor of the number of social distancing violations ($b = 1.98$, $p < 0.001$).

## Discussion

To slow down epidemic outbreaks, it is essential that people comply with social distancing directives [44]. However, it has proven difficult to assess the extent of people's compliance with such directives due to a lack of suitable data. To understand how members of the public comply with COVID-19 mitigation measures, recent studies have, for example, examined how compliance levels vary between and within countries [45] and how individual differences in

**Table 2. Number of social distancing violations regressed on the number of people on the street and each of the other variables.**

| | Model 1 | Model 2 | Model 3 | Model 4 | Model 5 | Model 6 | Model 7 | Model 8 |
|---|---|---|---|---|---|---|---|---|
| Intercept | -0.00 | -0.00 | -0.00 | -0.00 | -0.00 | -0.00 | -0.00 | -0.00 |
| | (3.74) | (5.43) | (5.21) | (5.44) | (5.28) | (5.43) | (5.45) | (5.21) |
| People on the | 1.24*** | 1.74*** | 1.68*** | 1.83*** | 1.63*** | 1.81*** | 1.79*** | 1.64*** |
| street | (0.27) | (0.38) | (0.34) | (0.34) | (0.37) | (0.33) | (0.40) | (0.35) |
| Google | 1.98*** | | | | | | | |
| mobility | (0.48) | | | | | | | |
| Temperature | | 0.80 | | | | | | |
| | | (1.95) | | | | | | |
| COVID-19 | | | 0.21 | | | | | |
| deaths | | | (0.18) | | | | | |
| COVID-19 | | | | -0.01 | | | | |
| transmissions | | | | (0.03) | | | | |
| *Corona* | | | | | -0.22 | | | |
| media items | | | | | (0.22) | | | |
| *COVID-19* | | | | | | -0.66 | | |
| media items | | | | | | (1.69) | | |
| *Corona* | | | | | | | -0.05 | |
| Google scores | | | | | | | (0.49) | |
| *COVID-19* | | | | | | | | -0.54 |
| Google scores | | | | | | | | (0.46) |
| $R^2$ | 0.84 | 0.66 | 0.69 | 0.66 | 0.68 | 0.66 | 0.66 | 0.69 |
| N | 18 | 18 | 18 | 18 | 18 | 18 | 18 | 18 |

***$p < 0.001$, **$p < 0.01$, *$p < 0.05$.

Estimates from trivariate OLS regression models. All variables were de-trended prior to including them in the models by taking residuals after OLS-regressing them on time, time squared, and periodicity due to weekday. Standard errors in parentheses.

compliance are related to personality, political orientation, and demographics [15–17]. What these studies have in common is a reliance on people's self-reports, which are limited by social desirability and recall bias. As such, prior work in this area captures people's *intentions* to comply with directives, rather than their actual compliance *behaviors*. Several attempts have been made to more objectively capture behavioral patterns using aggregated mobile phone location data [10–14]. This work has proven useful to assess community levels of compliance to stay-at-home directives, but these methods do not enable a fine-grained examination of the physical distance between people. Even though such physical distance is, ultimately, how social distancing measures help to prevent the virus from spreading [46]. Therefore, in the current study, novel techniques for systematic observation based on CCTV clips [47] have been applied to provide an objective overview of trends in people's physical proximity to others.

The study has four key findings. First, the study shows that people started to keep physical distance from each other in public spaces even before the social distancing directives were officially implemented. This is in line with recent work, which shows that during the COVID-19 pandemic, changes in economic activity [48] and adherence to stay-at-home directives [10] preceded formal regulations by the authorities. This speaks to the central function of voluntary action as opposed to governmental interference (e.g., school closures, general lockdowns) for the succes of non-pharmaceutical measures in reducing transmission rates [49, 50], at least during the first stages of a pandemic.

Second, the current study showed that, even though people complied with the social distancing directives prior to and immediately after these were first implemented, compliance started to decline gradually in the following weeks. This decline occurred despite repeated calls from government officials to continue keeping distance. The observed pattern of declining compliance to social distancing directives is in line with findings from other studies on the COVID-19 pandemic [8, 21] and mirrors patterns observed in studies on medical adherence outside the pandemic context [23, 51], which found that rapid declines in compliance occurred even in the first five to ten days of treatment [52, 53]. In a report in 2003, the World Health Organization concludes that a substantial number of patients do not adhere to their doctors' directives and that such poor adherence increases with the duration of the treatment regiments. Patterns of poor adherence were found across a range of conditions, including life-threatening diseases such as diabetes and HIV/AIDS [23]. Thus, in general, people tend to find it challenging to change their routines and lifestyles, even if it is paramount to their own health and well-being.

The third key finding concerns the strong correlation between the number of people on the street and the number of social distancing violations, which is in line with findings from at least one other study [8]. The correlation could be an indication of causality, such that the number of people on the street would affect people's willingness or their ability to keep distance. In narrow passage-ways (e.g., tunnels, bridges) and other spaces that impose physical constraints on people's movements, it might be more difficult for people to keep their distance when it is busy. In such locations, violation of social distancing directives may not (only) reflect people's willingness to comply, but their ability to succeed in doing so. It is unclear to what extent this opportunity argument is applicable to the observed social distancing violations in the current study. The three selected locations were large open areas with a few bottle-necks around entrances of shops and public transportation facilities.

An alternative explanation for the established correlation could be that the number of people on the street has a psychological effect on people's behavior. Relatively empty streets in otherwise busy areas might serve as visual cues that something is wrong [24, 25]. Such disruptions in social order could remind people to stay alert and could make them more aware of their own behavior. When the setting turns back to normal as more people are out in public space, there are fewer visible warning signs to remind individuals to keep their distance. Again, it is unclear to which extent this argument applies to our findings. At each of the locations, there were on average 25 people present in the time-periods selected for observation.

The fourth key finding concerns the relationship between our indicator of compliance to stay-at-home directives (i.e., Google Mobility data on time spent at non-residential locations) and our indicator of compliance to keep-distance directives (i.e., violations as observed from the CCTV footage) that persisted after accounting for the number of people on the street. In other words, we found that on days at which more time was spent at non-residential locations, people were less inclined to keep their distance *irrespective* of the number of people on the street. The established relationship might reflect an underlying sentiment toward government regulations, which changes from day-to-day due to external events such as riots, inspirational community initiatives, and speeches from central political figures.

An important implication of our findings is that keep-distance directives may only work in combination with stay-at-home directives and with avoid-busy-places directives. It appears that when people do not comply with stay-at-home directives, they also do not comply with 1.5-meter distancing directives. This means that the physical distancing violations are likely to increase as soon as the government relaxes the preventive measures and allows for schools, restaurants, and similar venues to reopen. It also means that policy-makers cannot rely on people keeping the 1.5-meter distance from others in public while allowing for relaxation of stay-at-

home directives. The conclusions of the study provide support for policies that account for the areal surface and the temporal patterns in visitor density to establish the number of people who are allowed in a specific location. Examples of such place-specific crowd-control policies are the allowance of a maximum number of people in stores and the closure of public spaces (e.g., parks, playgrounds, shopping streets) at times that they are expected to become too crowded.

A related implication is that indicators of community-wide mobility (e.g., as captured with cell phone location data) and of the number of people on the street might be used as proxies to determine compliance with directives about keeping physical distance. Although our study did not address people's engagement in other protective behaviors (e.g., wearing facemasks, increasing frequency of handwashing), it is possible that indicators of community-wide mobility and the number of people on the street could form proxies for such behaviors as well. Research suggests that individuals who perceived one type of protective behavior to be effective (e.g., wearing facemasks, home disinfection) are more likely to engage in other types of protective behaviors [54]. This would be good news for policy-makers, because it is easier to determine when locations are too crowded than it is to establish whether people are violating specific directives such as keeping their 1.5-meter distance. Such a proxy would be useful, for example, to determine if it is time for the installment of additional measures to ensure compliance such as stricter sanctioning (e.g., fines, forced stay-at-home) or efforts to increase awareness among the public about advisable behavior [55, 56].

In addition to examining the role of the number of people on the street and community-wide mobility, we also looked into other factors that could potentially explain compliance to keep-distance directives; the transmission of COVID-19 in the Netherlands, the temperature, and collective attention to COVID-19 as indicated by internet search behavior and media attention. We did not establish any contemporaneous significant effects between these factors and the number of social distancing violations after controlling for the number of people on the street, illustrating again the importance of the latter factor for people to keep physical distance from each other.

That said, the limited number of observations ($N$ = 18) will likely have restricted our ability to detect effects of small to moderate effect sizes. Also, both proxy measures of collective attention are limited to some extent. The necessity to search for information about a virus on the internet may decrease as its' spread progresses through the community, and as people become informed through media or individual experience. The media data only provide insight into how many news messages were produced and not into how many of those were read. An alternative way to operationalize collective attention could be through survey research, questioning nationally representative samples on multiple occasions about cohesion and trust [17]. Also, more details are warranted on the affective language in the media messages to assess, for example, whether the messages reflect skepticism toward social distancing guidelines. This could involve a discourse analysis of media content [57]. Finally, the data in the current study are not suitable to address the motivations behind individuals' decision-making, even though these are likely predictors of individuals' behavioral compliance. Survey studies suggest that people's willingness to comply with social distancing directives is associated with, for example, their perception of the disease as a threat, their perception of other people's compliance, and their trust in the authorities [15–17].

Replication of this study in a larger area, in an international context, and over more extended periods is warranted. Our findings are based on a study in the city of Amsterdam in the Netherlands. It is possible that the findings have a limited generalizability to other contexts owing to, for example, national differences in attitudes toward government intervention [58, 59] or cultural differences in personal space boundaries [60]. Also, other countries have

adopted different approaches in their social distancing policies and decided on distances of, for example, 1 meter (Austria), 1.8 meters (6 feet; USA), or 2 meters (Denmark). There are also substantial international differences in penalties for violations. Whereas violations in some countries are punished with fines (EUR 390 in the Netherlands during the study period), other countries have presented the social distancing directives as 'adviced behavior' rather than as regulation (e.g., Sweden). Due to the labor-intensive nature of systematic observation, we limited the period of our study to the first weeks after the implementation of the social distancing directives. It is unclear if the observed increase in non-compliance will persist in the post-events of the COVID-19 outbreak. Further, we selected cameras, days, and timeslots for coding rather than coding all available footage. We selected cameras at busy locations where the risk of social distancing directives is arguably highest, which may have inflated the violation rate. To enable the collection and processing of data about physical proximity between people on a larger scale, promising technologies are ultra-wideband, Bluetooth, pedestrian tracking sensors, and machine learning [8, 9, 61, 62].

Finally, we note that the current study concentrates on one aspect of social distancing: keeping physical distance from strangers in public places. Other measures, such as preventive quarantine, proactive tracing of potential positive cases, and the protection of professionals exposed to the public, will also aid in mitigating the spread of contagious diseases [20]. The assessment of compliance with these measures is beyond the scope of the current study, but also in dire need of exploration.

## Conclusion

The current study applied a novel approach of analyzing CCTV clips to assess people's behavior in public spaces. We conclude that compliance with social distancing directives appears to decline in the weeks after initial implementation. We further established strong correlations between, on the one hand, the number of observed physical distancing violations and, on the other hand, the number of people on the street and community-wide mobility. The findings imply that directives about keeping distance may work best in combination with stay-at-home directives and place-specific crowd-control strategies and, as such, policy-makers should not rely on people keeping the 1.5-meter distance from others in public while allowing for relaxation of stay-at-home directives. The findings also imply that indicators of community-wide mobility (such as captured with cell phone data) and of the number of people on the street might be used as proxies to assess whether people keep sufficient physical distance from each other at specific times and locations.

## Supporting information

**S1 Appendix. Codebook.**
(PDF)

**S2 Appendix. Coded violations and number of people on the street with unimputed data.**
(PDF)

**S3 Appendix. Time series analyses.**
(PDF)

**S4 Appendix. Bivariate correlations and trivariate regression models with raw data.**
(PDF)

**S5 Appendix. Detrending procedure.**
(PDF)

## Acknowledgments

We would like to thank Lisa van Reemst and Josephine Thomas for being involved in data descriptions, and the Amsterdam Police and Municipality of Amsterdam for facilitating the data collection. We are particularly grateful to Maikel van Scheppingen.We would also like to thank the Algemeen Nederlands Persbureau (ANP) for sharing their data on the number of COVID-19 related media messages.

## Author Contributions

**Conceptualization:** Evelien M. Hoeben, Wim Bernasco, Lasse Suonperä Liebst, Marie Rosenkrantz Lindegaard.

**Data curation:** Wim Bernasco, Carlijn van Baak, Marie Rosenkrantz Lindegaard.

**Formal analysis:** Wim Bernasco.

**Funding acquisition:** Marie Rosenkrantz Lindegaard.

**Investigation:** Evelien M. Hoeben, Wim Bernasco, Carlijn van Baak, Marie Rosenkrantz Lindegaard.

**Methodology:** Evelien M. Hoeben, Wim Bernasco, Lasse Suonperä Liebst, Marie Rosenkrantz Lindegaard.

**Project administration:** Evelien M. Hoeben, Marie Rosenkrantz Lindegaard.

**Resources:** Evelien M. Hoeben, Wim Bernasco, Marie Rosenkrantz Lindegaard.

**Software:** Wim Bernasco.

**Supervision:** Marie Rosenkrantz Lindegaard.

**Validation:** Evelien M. Hoeben, Wim Bernasco, Lasse Suonperä Liebst.

**Visualization:** Evelien M. Hoeben, Wim Bernasco.

**Writing – original draft:** Evelien M. Hoeben, Lasse Suonperä Liebst.

**Writing – review & editing:** Evelien M. Hoeben, Wim Bernasco, Lasse Suonperä Liebst, Carlijn van Baak, Marie Rosenkrantz Lindegaard.

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
