## [Decision Letter · Decision Letter 0]

1 Aug 2020

PONE-D-20-19629

Social distancing compliance: A video observational analysis

PLOS ONE

Dear Dr. Hoeben,

Thank you for submitting your manuscript to PLOS ONE. After careful consideration, we feel that it has merit but does not fully meet PLOS ONE’s publication criteria as it currently stands. Therefore, we invite you to submit a revised version of the manuscript that addresses the points raised during the review process.

Both of the reviewers have indicated concerns with the approach taken to analyse the data and the possible limitations in the data. It is important, that these concerns are addressed and if needed, a statistician sought to assist with updating the analysis.

We look forward to receiving your revised manuscript.

Kind regards,

Holly Seale

Academic Editor

PLOS ONE

Journal Requirements:

2.We note that you have indicated that data from this study are available upon request. PLOS only allows data to be available upon request if there are legal or ethical restrictions on sharing data publicly. For more information on unacceptable data access restrictions, please see http://journals.plos.org/plosone/s/data-availability#loc-unacceptable-data-access-restrictions.

Additional Editor Comments (if provided):

Both of the reviewers have indicated concerns with the approach taken to analyse the data and the possible limitations in the data. It is important, that these concerns are addressed and if needed, a statistician sought to assist with updating the analysis.

Reviewers' comments:

Reviewer's Responses to Questions

**Comments to the Author**

1. Is the manuscript technically sound, and do the data support the conclusions?

Reviewer #1: Partly

Reviewer #2: No

2. Has the statistical analysis been performed appropriately and rigorously? 

Reviewer #1: Yes

Reviewer #2: No

3. Have the authors made all data underlying the findings in their manuscript fully available?

Reviewer #1: Yes

Reviewer #2: Yes

4. Is the manuscript presented in an intelligible fashion and written in standard English?

Reviewer #1: Yes

Reviewer #2: Yes

5. Review Comments to the Author

Reviewer #1: Overall, I was excited to read this paper that uses video data to measure social distancing compliance. This is an important source of data to complement the more prevalent uses of cell phone data.

I have the following concerns/comments:

1. The authors should clarify how 1.5m is measured in the videos. It is well known that a lot of media images showing beaches are misleading because of the focal points --- is this an issue here?

2. It looks like most of the change in 1.5m compliance occurred prior to the national mandate. This is consistent with previous work showing the importance of voluntary measures in the US. E.g.,

- Chetty, Raj, John N. Friedman, Nathaniel Hendren, and Michael Stepner. 2020. Real-Time Economics: A New Platform to Track the Impacts of COVID-19 on People, Businesses, and

Communities Using Private Sector Data.

- Allcott, Hunt, Levi Boxell, Jacob Conway, Billy Ferguson, Matthew Gentzkow, and Benny Goldman.

2020. Economic and Health Impacts of Social Distancing Policies during the Coronavirus Pandemic.

In general, it would also be good to make sure the literature review is up to date when published.

3. In a lot of the correlations, the authors detrend the data. I'm not sure that is what you want to do. E.g., When examining the relationship between covid cases and social distancing behaviors, most people are going to be making inferences about the trends --- not deviations from those trends. It would be good to visualize the correlations both ways.

4. I'd maybe suggest removing the time series analysis section and placing it in a supplementary appendix. It's hard to know what inferences to make from these. I'd be interested in seeing a multivariate regression analysis looking at what predicts violations (e.g., cases, deaths, temperature, media mentions, etc).

5. I'm a bit concerned that the rise in 1.5m violations is purely mechanically related to the number of people in the video. I think it is important to try to distinguish between (a) 1.5m violations increasing purely because the # of people out is increasing vs (b) 1.5 m violations increasing because people are being less careful conditional on being outside. The correlation plot suggests a lot of this could be driven by (a).

Two ways I can think of addressing this:

- Compare the relationship between 1.5m violations and distancing in a placebo time peirod (e.g., a month or two before Covid or during the same period in 2019).

- Compute the expected number of violations if people were placed randomly in the video screen. (This isn't ideal, but could still be suggestive of how much of this is mechanical.)

Other points not required for revision, but would be great to see addressed if possible:

6. I was disappointed with the limited scope of the data actually analyzed. I was hoping machine learning was used to identify violations which would have allowed a minute-by-minute overview of these patterns across time and much richer analysis. The study as-is is still useful, but not to the same degree.

7. It would be great to compare the video violations to cell phone distancing behavior --- since that was part of the motivation for the paper.

8. It would be nice if the violations were also coded by demographics. E.g., Were 50% of all people on the video male, but males composed 70% of the violations?

Reviewer #2: This paper studies an important and timely question: examining whether citizens are adhering to social distancing practice set by the government, and whether such adherence last over time is critical to inform how we project the effectiveness of such policies. This study uses a novel measure of social distancing behaviors from CCTV camera footage in Amsterdam, a measure that is rare in the existing literature.

With this said, I think the paper’s analyses have a number of important limitations.

1. The authors do not perform any rigorous statistical tests, and hence it is impossible to tell whether there indeed is an upward trend of social distancing violation over time. With the current number of observations in the analyses, I do not think the authors have sufficient statistical power to draw any conclusions.

2. The primary outcome of interest is the total level of social distancing violation. This is very difficult to interpret, since as the authors show, the total number of people shown on the street is also increasing over time. Hence, it is not clear whether the rate of social distancing violation actually increases, which I believe is what the authors ultimately are interested in. Moreover, I do not think the current evidence can substantiate the claim that social distancing compliance and stay-at-home compliance coincide. One could mechanically result in the other: the absolute number of social distancing violation would go up (even if the compliance rate remains constant) if people are less likely to stay at home.

6. PLOS authors have the option to publish the peer review history of their article (what does this mean?). If published, this will include your full peer review and any attached files.

Reviewer #1: No

Reviewer #2: No

---

## [Author Response · Author response to Decision Letter 0]

12 Sep 2020

For our response to the reviewers, please see the enclosed document 'Response to Reviewers20-09-11'.

---

## [Decision Letter · Decision Letter 1]

4 Jan 2021

PONE-D-20-19629R1

Social distancing compliance: A video observational analysis

PLOS ONE

Dear Dr. Hoeben,

Thank you for submitting your manuscript to PLOS ONE. After careful consideration, we feel that it has merit but does not fully meet PLOS ONE’s publication criteria as it currently stands. Therefore, we invite you to submit a revised version of the manuscript that addresses the points raised during the review process.

Based on these reports, and my own assessment, I am pleased to inform you that it is potentially acceptable for publication, however, it is critical that you carry out the essential revisions suggested by our reviewers. Two of the reviewers have raised concerns that their previous suggestions have not been adequately addressed. Can I please ask that you ensure that you consider each of the suggested revisions and provide some dialogue around how the revision has been addressed (if appropriate). 

We look forward to receiving your revised manuscript.

Kind regards,

Holly Seale

Academic Editor

PLOS ONE

Reviewers' comments:

Reviewer's Responses to Questions

**Comments to the Author**

1. If the authors have adequately addressed your comments raised in a previous round of review and you feel that this manuscript is now acceptable for publication, you may indicate that here to bypass the “Comments to the Author” section, enter your conflict of interest statement in the “Confidential to Editor” section, and submit your "Accept" recommendation.

Reviewer #1: All comments have been addressed

Reviewer #2: (No Response)

Reviewer #3: (No Response)

2. Is the manuscript technically sound, and do the data support the conclusions?

Reviewer #1: Yes

Reviewer #2: No

Reviewer #3: Partly

3. Has the statistical analysis been performed appropriately and rigorously? 

Reviewer #1: Yes

Reviewer #2: No

Reviewer #3: Yes

4. Have the authors made all data underlying the findings in their manuscript fully available?

Reviewer #1: Yes

Reviewer #2: (No Response)

Reviewer #3: No

5. Is the manuscript presented in an intelligible fashion and written in standard English?

Reviewer #1: Yes

Reviewer #2: (No Response)

Reviewer #3: Yes

6. Review Comments to the Author

Reviewer #1: I found the revised manuscript more suitable for publication. I view the main contribution to be demonstrated in Figure 10 and Table 2--- the correlation between 1.5m violations, people on the street, mobility patterns, and covid-19 searches. It is important to show the extent to which different measures of social distancing are correlated, and this paper provides new evidence on this for 1.5m violations in public. I think this meets criteria for publication in PLOS One.

The manner in which the authors detrend each series separately is not standard procedure in the social science background I come from. Furthermore, if the main point is to show that different measures of social distancing are correlated, then you are interested in both the correlations in the trend and in the deviations from that trend. This isn't a causal question where you try to remove confounds, it is purely a descriptive question. For these reasons, I'd like to see, at least, a version of Figure 10 that shows the correlations of the raw data.

The authors could probably also shorten the discussion of some of the other aspects of the paper. I think less is more in this context given data limitations.

My only other comment is that there are a lot of figures in this paper. The authors could probably combine the plots with the violations on the Y-axis and the covariats on the x-axis into a single figure with multiple panels. And then a similar plot with the covariates on the Y-axis and the date on the x-axis into a single figure with multiple panels (possibly in the appendix).

Reviewer #2: I do not think the author has satisfyingly addressed my second comment from the previous review. In particular, the author should put "people on the street" used in Table 2, or just simply counting the number of people on the street from the CCTV camera stream, on the RHS in Table 1. Without doing so, the conclusion "directives about keeping distance may work best in combination with stay directives" is not substantiated, as it is impossible to tell whether the lack of social distancing on the street is simply due to an increase in total number of people showing up on the street in the first place.

Reviewer #3: I enjoyed reading this well-written article and the use of novel data sources to study human behavior. As the authors point out, manually coding hours of video footage can be labor intensive, which might limit the full potential that such data source represents. I would also argue that having two researchers to watch and manually code hours of video footage increases the risk of biases and errors. Have the authors considered using a machine learning approach instead? Alternatively, could the authors employ incentivized subjects (e.g. MTurk or students) to review the footage? This could help minimize errors and perhaps also use longer hours of footage (increasing the sample size).

The authors also use new deaths and infections as a control. Would the same results hold if the authors only used deaths and cases within a smaller Km radius around the footage locations? Also, are deaths and cases considered as a percentage of the population? Another interesting test could be to use new COVID-19 cases between the previous week and the week of the footage, which might help control for saliency of infections.

The authors also argue that their data source is more reliable than other studies using mobile data because they can better assess social proximity. However, most studies that use mobile data can measure proximity, for instance generating a measure of gyration (see Pepe et al. 2020, Scientific Data 7, 230). If the data allows, the authors could consider creating similar index of proximity and compare it with their footage data. If this is not possible, the authors should at least revise such statements from the manuscript.

Can the authors add some screenshots of the footage in an Appendix, perhaps blurring faces to preserve anonymity of passengers? This would help the readers contextualize how different the streets looked between crowded and less crowded days. Also, from the footage, can the authors see whether or not passengers were wearing a mask? Was there any notable change in mask-waring behaviors over time and between mask-wearing and distancing?

The authors should support with further tests and regressions their choice of the polynomial function, especially to test the null hypothesis of linearity, against the alternative that the regression is quadratic and/or cubic (e.g. a simple t-test should suffice). This can help control for whether differences between models are partly driven by outliers, which can be a problem in quadratic regression functions.

7. PLOS authors have the option to publish the peer review history of their article (what does this mean?). If published, this will include your full peer review and any attached files.

Reviewer #1: No

Reviewer #2: No

Reviewer #3: No

---

## [Author Response · Author response to Decision Letter 1]

15 Feb 2021

*This text was copy-pasted from the document 'Response to Reviewers', which is also enclosed*

Dear Editor,

We thank you for the opportunity to revise and resubmit our manuscript. We value the reviewers’ feedback and appreciate the time they have spent on reading our work.

In the remainder of this memo, we explain how we have addressed the suggested revisions. 

With kind regards, 

The authors 

Comments from the Editor

Based on these reports, and my own assessment, I am pleased to inform you that it is potentially acceptable for publication, however, it is critical that you carry out the essential revisions suggested by our reviewers. Two of the reviewers have raised concerns that their previous suggestions have not been adequately addressed. Can I please ask that you ensure that you consider each of the suggested revisions and provide some dialogue around how the revision has been addressed (if appropriate). 

In response to the reviewers’ comments, we have conducted additional analyses and a new literature search, added Appendices S4 and S5, and substantially shortened the discussion of the results. 

Comments from Reviewer #1 

The manner in which the authors detrend each series separately is not standard procedure in the social science background I come from. Furthermore, if the main point is to show that different measures of social distancing are correlated, then you are interested in both the correlations in the trend and in the deviations from that trend. This isn't a causal question where you try to remove confounds, it is purely a descriptive question. For these reasons, I'd like to see, at least, a version of Figure 10 that shows the correlations of the raw data.

We have calculated the bivariate correlations (Fig 6, which was Fig 10 in the previous version) and trivariate regression models (Table 1 and 2) with raw data, and added these as supplemental materials (Appendix S4). We refer to these materials on page 12, line 254.

The authors could probably also shorten the discussion of some of the other aspects of the paper. I think less is more in this context given data limitations.

We have substantially shortened the discussion of the results with respect to the covariates temperature, media items, and Google search scores.

My only other comment is that there are a lot of figures in this paper. The authors could probably combine the plots with the violations on the Y-axis and the covariates on the x-axis into a single figure with multiple panels. And then a similar plot with the covariates on the Y-axis and the date on the x-axis into a single figure with multiple panels (possibly in the appendix).

We followed the suggestion of the reviewer by combining the left panels of former Figures 1, 5, 6, 7, 8, and 9 into the new Figure 1 and the right panels of these former Figures into the new Figure 5. 

Comments from Reviewer #2 

I do not think the author has satisfyingly addressed my second comment from the previous review. In particular, the author should put "people on the street" used in Table 2, or just simply counting the number of people on the street from the CCTV camera stream, on the RHS in Table 1. Without doing so, the conclusion "directives about keeping distance may work best in combination with stay directives" is not substantiated, as it is impossible to tell whether the lack of social distancing on the street is simply due to an increase in total number of people showing up on the street in the first place.

We do not share the view of Reviewer #2 that taking the ratio of the number of violations by the number of people on the street as the dependent variable would offer an improvement to our study. Most importantly because it would not allow us to examine the effect of the number of people on the street on the number of violations. As we show with the analyses presented in Table 2, this is a non-trivial effect that explains away the effects of most but not all of the other covariates. 

As discussed on pages 18-19 of the paper (and Table 2), we find that people’s overall mobility, as captured with cellphone information from Google’s Community Mobility Reports, affects the number of social distancing violations even after correcting for the number of people on the street. This, in our opinion, substantiates our conclusion that directives about keeping distance may work best in combination with stay-at-home directives.

Comments from Reviewer #3 

Have the authors considered using a machine learning approach instead? Alternatively, could the authors employ incentivized subjects (e.g. MTurk or students) to review the footage? This could help minimize errors and perhaps also use longer hours of footage (increasing the sample size).

We agree with the reviewer that a machine learning approach would offer a time-efficient way of analyzing CCTV footage. We explicitly mention this as a promising venue for future research (page 26, line 548). In fact, we are currently working on a project to apply machine learning to automatically detect social distance violations from CCTV footage. However, this is a comprehensive undertaking that we consider to be beyond the scope of the current study. 

The authors also use new deaths and infections as a control. Would the same results hold if the authors only used deaths and cases within a smaller Km radius around the footage locations? Also, are deaths and cases considered as a percentage of the population?

The COVID-19 transmissions and deaths are included as absolute numbers. We added text to page 9, line 203 to make this explicit. If the population is stable over the examined period, which we expect it to be during such a small time-window (Feb 29th to May 2nd; it is not likely that a substantial number of people will die, be born, move in, or move away in that time-frame), it will not matter whether we include absolute numbers or percentages. 

We feel that our current approach, in which we use national-level data on COVID-19 transmissions and deaths, is preferred over including regional-level data. We theorized that individuals would be affected in their distancing behavior by the perceived urgency of the problem, as operationalized by the number of transmissions and deaths. Of course, people will only be affected in their behavior to the extent that they are aware of these numbers of transmissions and deaths. The Dutch media published these numbers at a national level. Therefore, it makes more sense to include national-level information on transmissions and deaths than information on transmissions and deaths that occurred at a small radius around the footage locations.

Another interesting test could be to use new COVID-19 cases between the previous week and the week of the footage, which might help control for saliency of infections.

We agree with the reviewer that this would be an interesting test. In the supplemental materials (Appendix S3), we present the cross-correlation function (CCF) between the number of violations on the one hand and the other variables (including new COVID-19 transmissions) on the other hand. Figure S3.2D shows that this correlation is slightly stronger negative in the two weeks prior (lag = -1 and lag = -2), compared to in the week itself (lag = 0). This means that the number of violations is slightly stronger associated with the number of new COVID-19 transmissions in the prior weeks than with the number of transmissions in the same week. We see a similar pattern for the number of COVID-19 deaths (Fig S3.2C).

The authors also argue that their data source is more reliable than other studies using mobile data because they can better assess social proximity. However, most studies that use mobile data can measure proximity, for instance generating a measure of gyration (see Pepe et al. 2020, Scientific Data 7, 230). If the data allows, the authors could consider creating similar index of proximity and compare it with their footage data. If this is not possible, the authors should at least revise such statements from the manuscript.

It is correct that other studies have measured proximity, however, not at the level of detail that our study does. To take the study of Pepe et al. (2020) as an example, they determine the location of respondents with a 10-meter accuracy margin (p.2) and respondents’ proximity to others based on 50-meter radiuses (p.3). Pepe et al. (2020, p.4) even explicitly state that “It is important to remark that this is not a close-range contact network.” By using video images, we can assess whether people are within 1.5 meter of each other, which is far more detailed than what mobile data, such as applied in the study by Pepe et al. (2020), allow for. 

Given the rapid growth of COVID-19 related research, we conducted a new literature search to look for studies on time trends in social distancing violations, which apply methods that (theoretically) assess physical proximity with a comparable level of accuracy. We found two studies that were published after our previous resubmission on Sep 12 (Sun et al., 2020, on Sep 25, and Pouw et al., 2020, on Oct 29). Sun et al. (2020) found that, among 1062 respondents from five European countries, the number of detected nearby Bluetooth-enabled devices was significantly lower during the lockdown compared to in the pre-lockdown period. Note, however, that the study does not specify within what range these devices could be detected. Pouw et al. (2020) apply pedestrian tracking sensors on crowds in a large train station. Their method is able to assess physical proximity quite accurately. They found a pattern in social distancing violations that is similar to the one we found, in that people complied with the guidelines in the beginning of the outbreak, but that the number of violations increased soon after. 

We recognize the potential of Bluetooth and pedestrian tracking sensors as venues for future research (page 26, line 548) and have included references to the studies by Sun et al. (2020), Pouw et al. (2020) and Pepe et al. (2020). We stand by our argument that our data source offers a more “fine-grained examination of the physical distance between people” compared to the use of aggregated mobile phone location data (page 20, lines 427-428), but we did revise our statement on page 4 to make more explicit how our study contributes to existing studies using mobile phone data.

Can the authors add some screenshots of the footage in an Appendix, perhaps blurring faces to preserve anonymity of passengers? This would help the readers contextualize how different the streets looked between crowded and less crowded days. 

This will not be possible due to the conditions under which the data were provided to us. Access to the CCTV footage data was provided by the Amsterdam police and municipality under the condition that data would be securely stored, not be publicly shared, and that the identity of the individuals visible in the footage would be protected. Even if we would blurr their faces, depicted individuals might be identified through their clothes and their presence at a specific location and time.

Also, from the footage, can the authors see whether or not passengers were wearing a mask? Was there any notable change in mask-waring behaviors over time and between mask-wearing and distancing?

During the time period covered in our study (Feb 29th-May 2nd 2020), facemask wearing was not common in The Netherlands, so few (if any) of the depicted individuals were wearing them. Formal mandates to wear face-masks in The Netherlands were not issued until August 2020 (regions Amsterdam and Rotterdam) and December 2020 (entire country).

The authors should support with further tests and regressions their choice of the polynomial function, especially to test the null hypothesis of linearity, against the alternative that the regression is quadratic and/or cubic (e.g. a simple t-test should suffice). This can help control for whether differences between models are partly driven by outliers, which can be a problem in quadratic regression functions.

We elaborated on the details of the detrending procedure in the main text (pages 11-12) and in the supplemental materials (Appendix S5). Specifically, we now present R2 and Bayesian Information Criterion measures as indicators of model fit as well as T-tests and likelihood ratio tests to compare each of the models against a more restricted version (also in Appendix S5). We explain how we decided on the polynomial function based on these measures.

---

## [Editor Report · Decision Letter 2]

23 Feb 2021

Social distancing compliance: A video observational analysis

PONE-D-20-19629R2

Dear Dr. Hoeben,

We’re pleased to inform you that your manuscript has been judged scientifically suitable for publication and will be formally accepted for publication once it meets all outstanding technical requirements.

Kind regards,

Holly Seale

Academic Editor

PLOS ONE
---

## [Editor Report · Acceptance letter]

5 Mar 2021

PONE-D-20-19629R2 

Social distancing compliance: A video observational analysis 

Dear Dr. Hoeben:

I'm pleased to inform you that your manuscript has been deemed suitable for publication in PLOS ONE. Congratulations! Your manuscript is now with our production department. 

Kind regards, 

on behalf of

Dr. Holly Seale 

Academic Editor

PLOS ONE